# Multi-Terminal Transistor-Like Devices Based on Strongly Correlated Metallic Oxides for Neuromorphic Applications

**DOI:** 10.3390/ma13020281

**Published:** 2020-01-08

**Authors:** Alejandro Fernández-Rodríguez, Jordi Alcalà, Jordi Suñe, Narcis Mestres, Anna Palau

**Affiliations:** 1Institut de Ciència de Materials de Barcelona, ICMAB-CSIC, Campus UAB, 08193 Bellaterra, Barcelona, Spain; afernandez3@icmab.es (A.F.-R.); jalcala@icmab.es (J.A.); narcis@icmab.es (N.M.); 2Departament d’Enginyeria Electrònica, Universitat Autònoma de Barcelona, 08193 Bellaterra, Barcelona, Spain

**Keywords:** strongly correlated oxides, resistive switching, neuromorphic computing, transistor-like devices

## Abstract

Memristive devices are attracting a great attention for memory, logic, neural networks, and sensing applications due to their simple structure, high density integration, low-power consumption, and fast operation. In particular, multi-terminal structures controlled by active gates, able to process and manipulate information in parallel, would certainly provide novel concepts for neuromorphic systems. In this way, transistor-based synaptic devices may be designed, where the synaptic weight in the postsynaptic membrane is encoded in a source-drain channel and modified by presynaptic terminals (gates). In this work, we show the potential of reversible field-induced metal-insulator transition (MIT) in strongly correlated metallic oxides for the design of robust and flexible multi-terminal memristive transistor-like devices. We have studied different structures patterned on YBa_2_Cu_3_O_7−δ_ films, which are able to display gate modulable non-volatile volume MIT, driven by field-induced oxygen diffusion within the system. The key advantage of these materials is the possibility to homogeneously tune the oxygen diffusion not only in a confined filament or interface, as observed in widely explored binary and complex oxides, but also in the whole material volume. Another important advantage of correlated oxides with respect to devices based on conducting filaments is the significant reduction of cycle-to-cycle and device-to-device variations. In this work, we show several device configurations in which the lateral conduction between a drain-source channel (synaptic weight) is effectively controlled by active gate-tunable volume resistance changes, thus providing the basis for the design of robust and flexible transistor-based artificial synapses.

## 1. Introduction

Digital computers can process a large amount of data with high precision and speed. However, compared to the brain, the computer still cannot approach a comparable performance considering cognitive functions such as perception, recognition, and memory. Neuromorphic computing, operating with a parallel architecture connecting low-power computing elements (neurons) with multiple adaptive memory elements (synapses), appears as a very attractive alternative to von-Neuman based algorithms in future cognitive computers [1,2]. The advantages of using analogue with very large-scale integration include: Inherent parallelism, as well as reducing the chip area and power consumption in comparison with digital implementations [3]. Design of computational systems mimicking the way that brain works, with intrinsically massive parallel information processing, is completely unfeasible by using the existing hardware which is based on conventional digital logic. Although stable learning has been achieved with digital logic for low-precision applications using binary weights [4,5], the development of novel functional materials, and individual device components able to resemble the properties of neurons and synapses, are mandatory to bring a revolutionary technological leap toward the implementation of a fully neuromorphic computer.

Resistive-switching devices, modeled as memristors, have become a leading candidate to mimic basic functionalities of biological components in a neural network, while providing clear advantages in energy and scalability [3,6]. The resistive switching effect consists of a non-volatile reversible switch between different resistance states, induced by an electric field [7]. Strongly correlated metal oxides showing metal–Mott insulating transitions (MIT) appear as particularly interesting materials for future neuromorphic device architectures, because they show large resistance variations, induced by small carrier concentration modulations, driven by an electric field, allotted to obtain multilevel analogue states [8,9,10]. The ability to continuously tune the electrical resistance, as well as to induce both volatile and non-volatile transitions, put them in a unique position to mimic neurons and synapses on a device level [11,12].

In particular, to achieve useful synaptic plasticity, a multistate behavior should be changed in an analog continuous fashion with long retention time so that the device resistance continuously depends on the electrical history. Spike time dependent plasticity (STDP) has been successfully demonstrated in different memristive devices based on two-terminal (2T) metal-insulator-metal passive circuit elements [13,14]. However, in biological systems, signal transmission and synapse learning are both generally regarded to occur concurrently in synapse-connected neuron pairs. Current 2T artificial synaptic devices operate by separating the signal transmission and self-learning processes in time. In this context, three-terminal (3T) synaptic devices, being able to realize both functions simultaneously, offer a promising solution for efficient synapse simulation [11,15,16]. Another reason why research on multiterminal devices is relevant is the possibility of having several gates which can obtain signals from different sources simultaneously, and they can therefore experience spatiotemporal effects, which 2T devices cannot [15]. Lately, new multiterminal devices that are able to mimic important aspects of biological sensing functions have been developed. In this sense, through the utilization of a simple organic electrochemical transistor based device with multiple gates, a sensing system has been demonstrated that is analogous to the orientation selectivity from the thalamus (the center part of the brain) to the visual cortex, which governs the vision process in the brain [17]. In a second example, a series of split-gate molybdenum sulfide transistors were implemented to mimic the coincidence nerve network in the owl’s brain [18]. Given the huge number of neurons and synaptic connections in the human brain, multi-terminal memristors are also needed to perform complex functions as heterosynaptic plasticity [19,20,21,22].

We have recently demonstrated stable volume field-induced resistive switching in structures based on strongly correlated metallic perovskite oxides (La_1−x_Sr_x_MnO_3_ (LSMO) and YBa_2_Cu_3_O_7-δ_ (YBCO)), modulated through oxygen diffusion [23,24]. Optimally doped LSMO and YBCO materials are metallic in its initial state and they evolve into the insulating state by decreasing the oxygen content [25,26]. Multiple resistance states, needed for synaptic applications, can be achieved by tuning the oxygen doping with the applied voltage. In order to elucidate this behavior, Figure 1 displays different curves obtained by measuring the *R*(*T*) evolution of a switched contact in a YBCO film, after applying different voltage pulses. A clear MIT transition from the optimally doped metallic state to an underdoped insulating state is observed.

The key advantage of these systems is that the resistive switching is based on the MIT, which being an intrinsic property of the material, causes a homogeneous change of resistance in a gate modulable volume, allowing the design of flexible transistor-like devices (memristors) [23]. It is worth noting that the volume resistance modulation, observed in metallic perovskites, offers enhanced robustness, in terms of cycle-to-cycle and device-to-device variations, when compared with that induced in strongly correlated oxides that are insulating in the pristine state, where the switching phenomena is strongly localized at the contact interface or in confined filaments [7,27].

Here, we report on the study of the oxygen diffusion in YBCO based multi-terminal memristor devices in which the oxygen redistribution, and thus the conductance of a drain-source channel, may be tuned by using various gates. A sketch of the oxygen diffusion mechanism occurring below the gate, emulating a synaptic process, is shown in Figure 2.

The conductance between a source-drain channel (post-synaptic membrane) is controlled through modulatory gate terminals (pre-synaptic inputs). By the application of a gate voltage, oxygen vacancies are redistributed within the YBCO channel, locally changing its doping level, thereby their resistance (conductance). Synaptic plasticity characteristics may be obtained with intermediate synaptic weight states achieved by tuning the amount of oxygen vacancies created. Multiple pre-synaptic input terminals have been emulated by using multiple intermediate gates between the drain-source channel.

## 2. Materials and Methods

The geometry of YBCO transistor-like devices consist of a drain-source channel with different gate terminals to modulate the channel conductance. Optimally doped, epitaxial YBa_2_Cu_3_O_7−δ_ (YBCO) thin films, with thickness of 100 nm, were grown by pulsed laser deposition (PLD) on (001)-LaAlO_3_ single crystal substrates. The parameters used in this process were previously optimized for our purposes. The substrate was heated up to T = 800–810 °C, with an O_2_ partial pressure of 0.3 mbar during the deposition and a fixed target-substrate distance of 52.5 mm. A high fluence laser (∼2 J/cm^2^) working at a frequency of 5 Hz was used. During the cooling ramp, we increase the P(O_2_) in the chamber in order to obtain well oxygenated samples. The thickness of the film is mainly determined by the number of pulses. For these samples, 2600 pulses were applied obtaining a thickness of 100 nm. Topography shows a high-quality flat surface in all cases, the root-mean-square (rms) value of surface roughness is found to be below 1 nm. The structural features of YBCO films have been studied by theta-2theta X-ray diffraction (Siemens Diffractometer D5000, Siemens AG, Munich, Germany). The epitaxial nature of the films was evidenced by the detection of only (001) peaks along with the corresponding (001) peaks originating from the (001)-LAO substrates in the theta-2theta X-ray diffraction spectra. Photolithography and wet etching were used to pattern channels with different widths, *w* = 5–100 μm. After the patterning, multiple 50 nm thick, 100 × 100 μm^2^ silver contacts, spaced different distances apart, *d* = 100–300 μm, were deposited by sputtering and lift-off.

Electrical measurements were performed at room temperature with a Keithley 2450 source-meter at ambient pressure and temperature. Voltage pulses of 4 s, were applied between two top gates (top–top configuration), located at different positions of the channel, while measuring the current–voltage (*I*–*V*), and associated resistance–voltage (*R*–*V*) characteristics, in a two-point configuration. The variation of the drain-source conductance through the channel, obtained after applying the gate pulses, was evaluated by measuring the resistance at intermediate segments of the channel, *N*, (*R_N_*, *N =* 1, 2, 3, 4, 5) in a standard four-point method, using two external electrodes to inject the current and intermediate contacts to measure the voltage. In this way, we avoid the contribution of the contact resistance and thus we obtain the bulk resistance change. Figure 3a shows a schematic representation of the proposed device and Figure 3b an optical microscopy image of several devices with different channel widths (*w* = 100, 50, 20, 10, and 5 μm).

## 3. Results and Discussion

### 3.1. Switching Characteristics between Two Gates

Figure 4 shows repeated *I*–*V* scans (Figure 4a), and the associated *R*–*V* curves (Figure 4b), obtained for a device with a channel of *w* = 50 μm, by applying positive and negative voltage pulses within two gate electrodes placed *d* = 200 μm apart. A complementary switching behavior was reproduced, since the two gates, see opposite voltage polarities in opposite directions [28,29]. Thus, for a given polarity, one electrode undergoes a set process (incorporating oxygen and thus switching from a high resistance state (HRS) to a low resistance state (LRS)) while in the other one a reset transition is produced (losing oxygen and switching from an LRS to an HRS). The set (*V_set_*) and reset (*V_reset_*) voltages, associated to each gate electrode, are depicted in Figure 4b. In general, for both polarities, *V_set_* occurs at lower voltage values than *V_reset_*, due to a fast motion of oxygen in the low conductivity regions [23]. In a device with symmetrical gates, the resistance values obtained at the HRS and LRS for a given voltage pulse are the same thus providing a symmetrical loop, as the one shown in Figure 4b.

The evolution of *V_set_* and *V_reset_* have been investigated by evaluating different *I*–*V* curves obtained for devices with different channel widths applying the minimum voltage pulse able to reversibly switch the gates at different distances (see Figure 5a). The values of *V_set_* and *V_reset_* increase linearly with the electrode distance, according to a constant dependence with the set and reset electric field (*E_set_* and *E_reset_*, respectively) which are of the order of, *E_set_* ~ (1–1.5) × 10^4^ V/m, *E_reset_* ~ (1.5–2) × 10^4^ V/m. Bias voltages of *V* ~ 2 V are needed to switch gates placed 10 μm apart and lower values, favorable for practical applications, are expected by reducing the device dimensions. Figure 5b shows the HRS and LRS resistances obtained for devices of different widths. The HRS have been read at *V* ~ 0 V, whereas for the LRS we considered the minimum value of the loop resistance. Both values are indicated in Figure 4b by dashed arrows.

It is clearly observed that the resistance values for the HRS and LRS increase with decreasing the channel width, with a behavior that is essentially linear, which is completely consistent with a volume resistive switching process [23]. Deviations of the linear dependence at low channel widths may be attributed to fabrication factors or border effects. In this way, by changing the gate area or unbalancing the applied positive/negative voltage pulse, one can modulate the weight of resistance variation in each gate. Figure 6a shows a typical example of a device with different gate areas in which the HRS and LRS of each gate exhibit different resistance changes, thus producing an asymmetrical *R*–*V* curve. Figure 6b shows an example of an asymmetrical *R*–*V* curve obtained in a device with equal gates but applying asymmetrical voltage pulses. In this case, the maximum applied negative voltage is lower than *V_reset_* and thus the contact that should switch at the HRS at this polarity does not change. Both situations will be better described in the following.

Next, we will demonstrate that the reversible gate resistance modulation, which occurs through a field-induced MIT driven by oxygen diffusion, is not just occurring below the gates, but also effectively modifies the volume resistance of the drain-source channel.

### 3.2. Conductance Modulation in a Drain-Source Channel

The conductance modulation of the device channels has been evaluated by measuring the relative variation of volume resistance at different segments of it (in a four-point configuration), after applying several positive and negative voltage pulses between two intermediate gates, in a two-point configuration. Figure 7a shows a schematic representation of the active gates (in yellow) and voltage probe positions considered for a device with a track of *w* = 10 μm and gate distance *d* = 100 μm. Figure 7b shows the *R*–*V* curves measured through the yellow gates by applying a maximum voltage of 12 V. A complementary switch of the two contact gates is clearly evidenced with a rather symmetrical *R*–*V* hysteresis curve. We plot in Figure 7c the percentage resistance change, measured at different segments of the channel, Δ*R_N_*, calculated by using Equation (1).
Δ*R_N_* = 100 × [*R_N_*(*t*) − *R_N_*(*i*)]/*R_N_*(*i*)(1)
were *R_N_*(*i*) and *R_N_*(*t*) are the resistance values measured at the *N* segment at the initial state and after several voltage pulses, respectively, using a four-point configuration. A clear correlation between ∆*R_N_* and the applied pulses is observed in Figure 7c. Large resistance variations (∆*R_N_* ~ 15–35%) are obtained at the segments close to the gates (*N* = 2, 3, 4), whereas the resistance does not change on those segments located further away from the gates (*N* =1, 5). It is worth pointing out that the variation of resistance at different regions of the channel compensate each other, providing a nearly constant resistance when measured through the whole channel (*R_TOTAL_*). This is in agreement with a redistribution of oxygen vacancies within the channel, with no external oxygen exchange, as modeled in [23].

The major effect of field induced oxygen diffusion is a drift of oxygen vacancies going from the negatively charged gate to the positive one. Thus, an accumulation of oxygen vacancies confined below the right gate occurs for positive voltage pulses (top Figure 7a), that is detected with a switching of this gate to the HRS in the two-point configuration measurement (Figure 7b). The complementary effect is obtained for negative voltage pulses with an accumulation of oxygen vacancies below the left gate (bottom Figure 7a) thus inducing a switching of this gate to the HRS. We have depicted this localized gate effect (from now on referred as gate switching) by coloring blue and red regions (not at scale) below the gates for LRS and HRS, respectively. The gate switching cycles produce a non-trivial reversible oxygen redistribution within the channel, that have been evaluated by measuring the resistance at different segments in a four-point configuration. The measured resistance at a given segment of the channel is directly correlated with its local oxygen concentration, being higher by increasing the amount of oxygen vacancies [25,30]. Thus, assuming a homogenous switch and considering that the resistivity of the HRS is much higher than that of the LRS, the amount of switched channel volume can be directly correlated from the resistance variation as schematically represented in Figure 7a. When one of the gate electrodes is switched to the LRS, there is a region nearby the gate that loses oxygen, i.e., segment *N* = 2 for the left gate (top Figure 7a) and segments *N* = 3 and *N* = 4 for the right (bottom Figure 7a). Although the device presents a rather symmetrical *R*–*V* curve (Figure 7b), the slightly different gate performance produces an asymmetry in the oxygen redistribution. That is, the right gate (with a higher HRS value) is able to inject oxygen vacancies in a wider lateral distance than the left one. It is worth noting that the oxygen vacancy accumulation/decrease produces a large effect in the oxygen redistribution within the channel that induces resistance changes not only in the segments localized between the gates but also in the concomitant ones, envisaging the large oxygen mobility occurring in these devices.

Conductivity modulation along the track can be tuned by further unbalancing the weight of the resistance switch in each gate. Figure 8 shows conductance measurements performed in a device with *w* = 100 μm and a gate distance *d* = 500 μm, considering the configuration schematically shown in Figure 8a. In this case the *R*–*V* hysteresis curves measured are clearly asymmetric, with very different high-resistance states achieved in the two gates (Figure 8b). The resistance variation at different segments of the channel for this device is shown in Figure 8c. In this case, the oxygen distribution is highly inhomogeneous, with a larger resistance change (~200%) concentrated close to the right gate (segment *N* = 4), and a complementary smoother resistance variation, that occupies a large region of the channel, nearby the left gate (segments *N* = 1, 2, and 3). As in the previous case, the gate that undergoes a transition to a more insulating state (left gate in this case) is the one that is able to inject oxygen vacancies to further lateral distances.

The maximum applied voltage in the *I*–*V* curves may also be used as a tuning knob to control the oxygen redistribution (and thus the conductance) through the channel. We show in Figure 9 an extreme case, for a device with *w* = 30 μm and *d* = 500 μm, in which we just switch one of the two gates, maintaining the other at the LRS. To do so, we apply a negative voltage pulse higher than *V_reset_* to switch the left contact to the HRS (Figure 9b). This contact is then switched back by to the LRS by applying a positive voltage higher than *V_set_* but lower than that needed to switch the right contact to the HRS (*V_reset_*).

The resistance variation through the channel is shown in Figure 9c and the associated oxygen redistribution is schematically depicted in Figure 9a. In this case, we observe that by switching the left gate to the LRS, oxygen vacancies are injected in the segments nearby (*N* = 1, 2). Subsequent voltage cycles produce a reversible motion of oxygen vacancies form *N* = 1, 2 to *N* = 3. Note that the conductance is not affected in the sections near the right gate, *N* = 4, 5, which is kept at the LRS.

Resistance experiments directly confirm that the conductance between a drain-source channel can be effectively modulated by using different active gates and that the oxygen redistribution in it strongly depends on the switching performance of each gate, and the applied voltage pulse. These results provide a proof-of-concept of resistance modulation in multi-gate memristor structures based on the strongly correlated materials showing the MIT. However, the top–top application of voltage strongly limits not only the device performance but also the complete characterization of synaptic functions. For practical applications, top–bottom configurations should be used to increase the on-off ratio, to allow reliable programming intermediate states and to design the required devices characteristics (linear conductance change, symmetric set and reset, retention, etc.). The proposed devices provide a wide design space based on material engineering (oxide doping, oxygen scavenging layers, etc.) and geometrical engineering (oxide thickness, separation between gates, number of gates, etc.) which may help to achieve the desired device characteristics for synapses and neurons.

## 4. Conclusions

Our work shows the potential of multi-terminal memristive structures, based on strongly correlated YBCO metallic oxide, as a promising approach for the design of neuromorphic devices, exploiting the tuneability of field-induced oxygen doping. Results demonstrate that multiple gates can be used to change the conductance (local oxygen doping) between a source-drain channel, thus emulating the synaptic weight. The movement and redistribution of oxygen vacancies within the channel, and thus its conductance, may be controlled by the device geometry, gate dimensions and position, and bias voltage. A large design flexibility can be obtained by changing the switching performance of different gates, thus offering the possibility to locally adjust the conductance response as required to implement neuromorphic functionalities.

## Figures and Tables

**Figure 1 materials-13-00281-f001:**
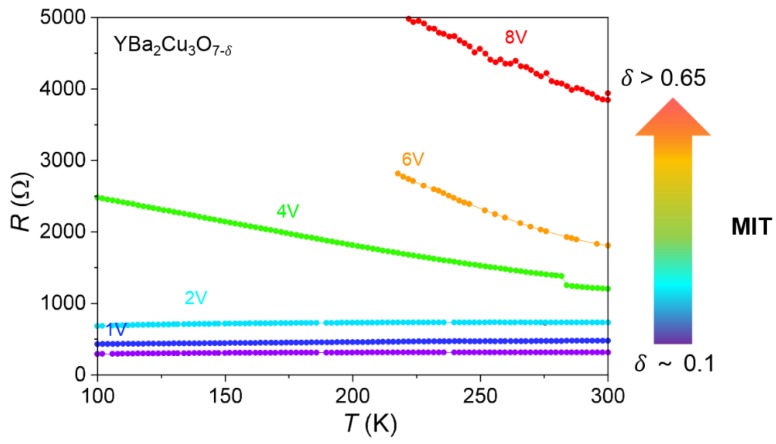
Resistance versus temperature obtained for a YBCO film after applying a series of voltage pulses to a silver contact of 100 μm.

**Figure 2 materials-13-00281-f002:**
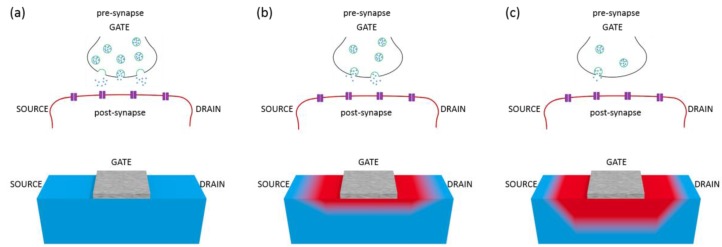
Schematic representation of a transistor-like device emulating a biological synapse. Blue and red in the bottom pictures depict optimally doped and under-doped YBCO, respectively. According to the oxygen doping, the schemes represent (**a**) high, (**b**) intermediate, and (**c**) low source-drain conductance.

**Figure 3 materials-13-00281-f003:**
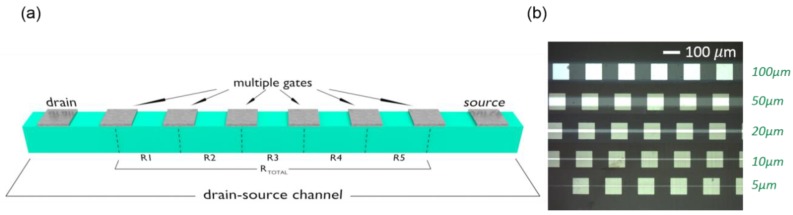
(**a**) Schematic representation of a transistor-like device with a source-drain channel and multiple tunable gates. The channel conductance is evaluated by measuring intermediate resistances (*R_N_*); (**b**) optical microscope image of several devices patterned with different channel widths from 5 to 100 μm.

**Figure 4 materials-13-00281-f004:**
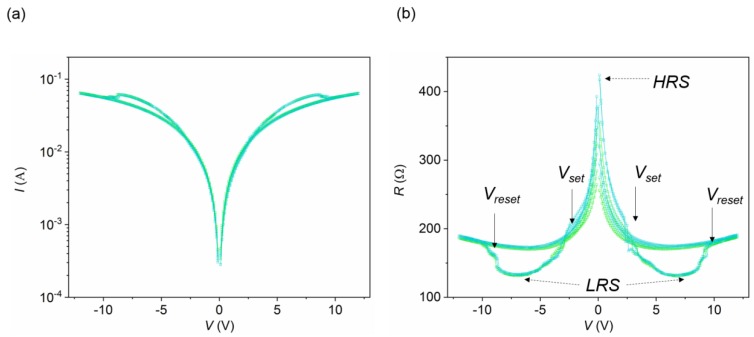
Typical (**a**) *I*–*V* and (**b**) *R*–*V*, obtained by using a two-point measurement, for a YBCO transistor-like device with a channel width of *w* = 50 μm, obtained by applying several voltage pulses through two identical gates separated at a distance of *d* = 200 μm, in a top–top electrode configuration.

**Figure 5 materials-13-00281-f005:**
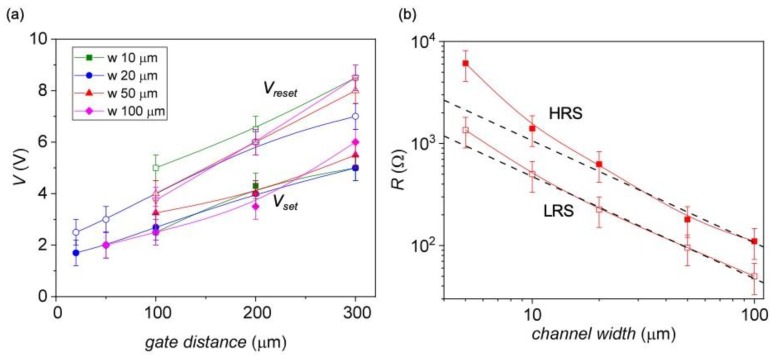
(**a**) Set (closed symbols) and reset (open symbols) voltages as a function of the gate distance obtained for devices of different widths; (**b**) evolution of the HRS and LRS resistance values with the device width. Dashed lines correspond to a linear dependence of R with the channel width, solid lines are guides to the eye. All values have been obtained by using a two-point configuration.

**Figure 6 materials-13-00281-f006:**
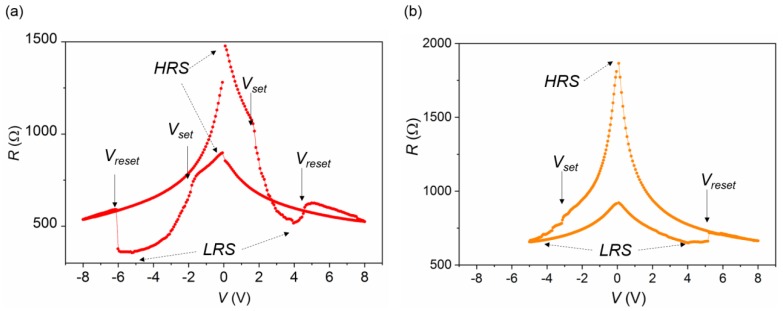
Typical *I*–*V* characteristics, obtained by using a two-point measurement, for YBCO transistor-like devices by applying (**a**) symmetrical voltage pulses using two gates with different switching performance; (**b**) asymmetrical voltages pulses to switch just one gate.

**Figure 7 materials-13-00281-f007:**
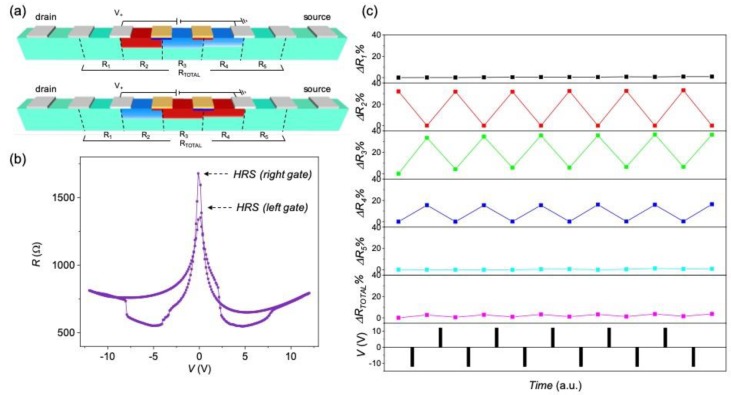
(**a**) Schematic representation of the oxygen redistribution in a YBCO device with a channel of *w* = 10 μm, by applying positive (top) and negative (bottom) voltage pulses between the two yellow gates separated at *d* = 100 μm. Red and blue colors represent HRS and LRS, respectively; (**b**) *R*–*V* characteristics, obtained by applying voltage pulses between the gates in two-point configuration; (**c**) percentage resistance change, measured at different segments of the device, using a four-point configuration, after a series of gate voltage pulses. The initial resistance of all segments was *R_N_* ~ 1500–2000 μm.

**Figure 8 materials-13-00281-f008:**
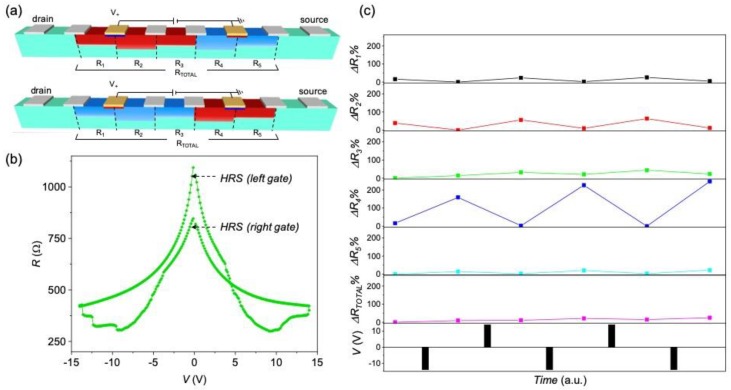
(**a**) Schematic representation of the oxygen redistribution in a YBCO device with a channel of *w* = 100 μm, by applying positive (top) and negative (bottom) voltage pulses between the two yellow gates separated at *d* = 500 μm. Red and blue colors represent HRS and LRS, respectively; (**b**) *R*–*V* characteristics, obtained by applying voltage pulses between the gates in two-point configuration; (**c**) percentage resistance change, measured at different segments of the device, using a four-point configuration, after a series of gate voltage pulses. The initial resistance of all segments was *R_N_* ~ 150–200 μm.

**Figure 9 materials-13-00281-f009:**
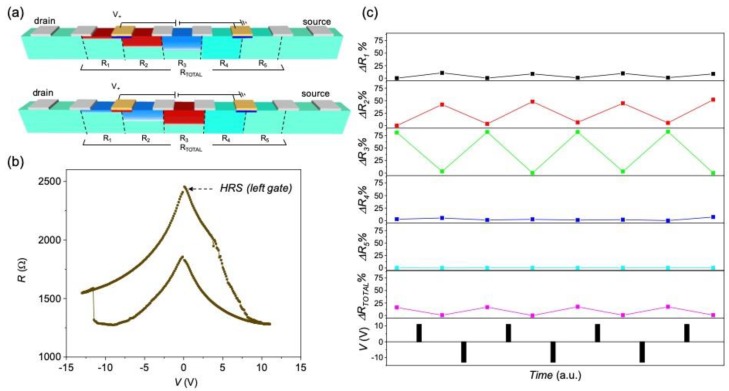
(**a**) Schematic representation of the oxygen redistribution in a YBCO device with a channel of *w* = 30 μm, by applying positive (top) and negative (bottom) voltage pulses between the two yellow gates separated at *d* = 500 μm. Red and blue colors represent HRS and LRS, respectively; (**b**) *R*–*V* characteristics, obtained by applying voltage pulses between the gates in two-point configuration. Maximum applied positive voltage pulse has been kept below *V_reset_* in order to maintain the right contact at the LRS; (**c**) percentage resistance change, measured at different segments of the device, using a four-point configuration, after a series of gate voltage pulses. The initial resistance of all segments was *R_N_* ~ 400–500 μm.

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
