# Peer review of "Multi-Terminal Transistor-Like Devices Based on Strongly Correlated Metallic Oxides for Neuromorphic Applications"

_materials, 2020, doi:10.3390/ma13020281_

Round 1

Reviewer 1 Report

While the author presented interesting findings, I am confused about the mechanisms. 1) It would be helpful to state how the resistance in each figure was made. Was it done using a two-point measurement, or a four-point measurement? Did the authors do any four-point measurements? 2) What is the relationship between resistance and gate distance? This would be the most straightforward way to assess whether switching is due to bulk vs contact resistance, but these results were not shown. 3) The mechanisms resulting in the red/blue spots in Fig.7-9 was unclear. The authors need to clearly explain which direction oxygen vacancies are moving when different voltages are applied. a) When a voltage is applied between two electrodes, which is the direction of oxygen vacancy movement with different signs? Does the electronic conductance increase with oxygen vacancy concentration, or decrease?  b) In Fig. 7, the authors claim that applying a positive voltage on gate #3 will attract oxygen vacancies into R3 from R2. I am confused how does that happen. If gate 2 is floating, why would there even be an electric field between gate 2 and 3 to induce oxygen vacancy movement from R2. I would only anticipate oxygen vacancy movement between the two gate electrodes where the contacts are made c) Why is the fifth gate in Fig. 9 always at the LRS? In FIg. 8, applying a positive voltage between gate 2 and 5 caused gate 5 to be colored red; however, the same experiment in Fig. 9 caused gate 5 to be blue. d) What do the colors on the gate represent? I understand the red/blue on the channel represents the conductivity, but if the mechanism is bulk switching, what does it mean for the gate to be LRS or HRS, and how was that determined? 4) What are the values of RN(i) for each segment of the device? The authors only show change but there’s no mention of absolute values.

Reviewer 2 Report

In this manuscript, the authors report on a multi-terminal transistor-like device based on  strongly-correlated metal oxides for neuromorphic computing applications. The manuscript is well written and the results are relatively clear. I therefore believe that the manuscript can be published in Materials. The authors may need to address the following issues in order to improve the quality of the manuscript:

At the introduction, the authors can refer to additional publications on multi-terminal devices and their ramifications in neuromorphic computing from a biological aspect (for example, Nature Communications volume 10, 3450 (2019), Scientific Reports 6, 27007 (2016)).  At the beginning of the introduction (line 38) the authors claim that "Design of computational systems mimicking the way that brain works, with intrinsically massive parallel information processing, is 39 completely unfeasible by using the existing hardware which is based on conventional digital logic." The authors can refer to advantages and disadvantages of digital vs analogue approach. For example digital logic can also be used for low precision applications in neuromorphics (for example binary synapses). Line 58 " However, 2T devices operate with separated signal transmission and learning process, while these two processes occur simultaneously in a biological synapse." The sentence is not clear, please rephrase.  I feel that beginning the manuscript with Figure 1 seems abrupt. The authors could consider to rearrange.  Fig. 5b. Is there any explenation based on device physics for the non-linear behaviour of R vs channel width?
